# Expression of genes with biomarker potential identified in skin from DSLD-affected horses increases with age

**Jennifer Hope Roberts[1], Jian Zhang[1], Florent David[2,3], Amy McLean[4], Karen Blumenshine[5], Eva Müller-Alander[6], Jaroslava Halper[1,7] ***

**1** Department of Pathology, College of Veterinary Medicine, University of Georgia, Athens, Georgia, United States of America, **2** Equine Care Group, Mazy, Gembloux, Belgium, **3** Equine Veterinary Medical Center–A member of Qatar Foundation, Doha, Qatar, **4** Department of Animal Science, College of Agricultural and Environmental Science, University of California at Davis, Davis, California, United States of America, **5** Santa Barbara Equine Practice, Santa Barbara, California, United States of America, **6** Pferdepraxis, Overath, Germany, **7** Basic Science Department, AU/UGA Medical Partnership, Athens, Georgia, United States of America

* jhalper@uga.edu

**Data Availability Statement:** All data are contained in Haythorn, A., et al., Differential gene expression in skin RNA of horses affected with degenerative suspensory ligament desmitis. Journal of

## Abstract

Degenerative Suspensory Ligament Desmitis (DSLD) negatively impacts connective tissues in horses, which often leads to progressive chronic pain and lameness. DSLD has been shown to be a systemic disorder that affects multiple body systems, including tendons, sclerae, and the aorta. Currently, the diagnosis is confirmed by post mortem histological examination of a tendon or suspensory ligament. Histology reveals inappropriate accumulations of proteoglycans in the tendons and other tissues in DSLD-affected horses. Unfortunately, there is no reliable method to diagnose DSLD in living horses. Recently, bone morphogenetic protein 2 (BMP2) was identified in active DSLD lesions. In addition, recent data from RNA sequencing (RNA-seq) showed overexpression of numerous genes, among them *BMP2*, *FOS* and genes for keratins in DSLD skin biopsies-derived RNA. We hypothesized that some of these genes can be used as biomarkers for diagnosis of DSLD in a panel. Overexpression of some of them was verified in quantitative real time PCR. Immunohistochemistry and RNAscope in-situ hybridization (ISH) assays were used to determine the level of overexpression of specific genes in skin biopsies from control and DSLD-affected horses. The RNAscope ISH assay has shown to be more reliable and more specific that immunohistochemistry. ISH confirmed a significant increase in *KRT83* and *BMP-2* in hair follicles in DSLD cases, as well as abnormally high expression of *FOS* in the epidermis, especially in aging horses. Because statistically relevant specificity and sensitivity was documented only for *FOS* and *BMP2*, but not *KRT83* we recommend the use of *FOS* and *BMP2* panel to diagnose DSLD. We conclude that a panel of two markers from the studied group (*BMP2* and *FOS*) can serve as an additional diagnostic tool for DSLD in living horses, especially in older animals. Further studies are necessary to confirm if this biomarker panel could be used as a prospective tool to identify DSLD in horses as they age.

Orthopaedic Surgery and Research, 2020. 15(1): p. 460. https://www.ncbi.nlm.nih.gov/Traces/study/?acc=PRJNA544650.

**Funding:** USDA Animal Health Capacity Grant to JH and JHR, private donations to JH, Global Research Collaborative Grant from University of Georgia to JH and FD, ORFGA Grant, Summer Research Stipends and Grants to JHR. Many thanks go to Foundation for Horses and Other Animals, Mr. and Mrs. Lou Gonda and El Campeon Farm for their generous support. The funders had no role in study design, data collection and analysis, decision to publish, or preparation of the manuscript.

**Competing interests:** The authors have declared that no competing interests exists.

## Introduction

Disorders that affect tendons and suspensory ligaments in horses are often chronic and debilitating. They are also frequently difficult to diagnose specifically because obtaining tendon sample for histological analysis carries a high degree of negative outcome due to biopsy-induced serious damage to tendon and/or ligament tissues (TLT) leading to poor healing and biomechanical impairment. The chance of a sustainable and favorable outcome is greatly diminished. Naturally-occurring flexor tendon and suspensory ligament injuries are notoriously challenging to heal in horses. Given months to years of recovery, the lesion site eventually heals with a variable amount of scar tissue, replacing the original healthy tissue. The scar tissue provides inferior biomechanical characteristics to the repaired tendon or ligament, often limiting the horse's performance significantly and making the horse prone to relapse. In this context, harvesting TLT biopsies is therefore contra-indicated, making additional iatrogenic tissue damage unethical in client-owned horses.

Degenerative suspensory ligament desmitis (DSLD) is one such disorder. DSLD is a puzzling ailment in horses that has been documented for decades, and yet no specific cause or cure has been established. It is typically characterized by usually bilateral dropped fetlocks or straightened hocks, as well as difficulty standing from a lying down position. In addition, there may be swelling of not only distal, but also of proximal limbs particularly in the area of the suspensory ligaments and its branches. Histological examination shows pathological changes not only in suspensory ligaments, but also in superficial and deep digital flexor tendons [1]. Eventually, as the disease progresses, the horses become lame and are often euthanized because of increasing pain. Owners also report that exercise and extensive conditioning of the horses does not inform on probability of DSLD development [2]. The disease can occur in any breed, age, or gender, but seems to be more prevalent in Peruvian Pasos, Warmblood breeds, Arabians, and Quarter Horses [1]. Similarly, in the study described here, most biopsies were obtained from Peruvian Pasos, followed by Arabians, Warmbloods and Hanoverians. Ultrasound examination of suspensory ligaments and/or tendons such as superficial digital flexor tendon (SDFT) and deep digital flexor tendon (DDFT) may reveal a diffuse loss of echogenicity and abnormal fiber patterns, especially in severe cases [2]. An increase in proteoglycan content has been documented in TLT, as well as in mediae of blood vessels such as the aorta, but this is only observed through post mortem histological examination. In addition, some owners report skin abnormalities, such as looseness and thinning, though no objective study has been done [3]. Currently, it is suspected that DSLD may be inherited, but no gene mutation or chromosomal aberration has been identified and linked directly to DSLD. Diagnosis of subclinical or apparently healthy horses is not feasible as no definitive identifiable cause (e.g., gene mutation, biochemical abnormality) has been documented. As mentioned above diagnosis can only be confirmed through a tendon biopsy and examination of the biopsy for proteoglycan accumulation, severe tissue organization disruption [4], and characteristic fibroblast/tenoblast whorls in initial stage of DSLD [5]. However, this approach would lead to further tendon/ligament impairment due to poor and incomplete healing of biopsied sites. Therefore, diagnostic methods that either involve tendons but are minimally or non- invasive, or are based on examination of other tissues than tendons need to be developed.

Our previous work, the only RNA-seq study done so far, showed that RNA-seq of skin RNAs from control and DSLD-affected horses showed not only marked overexpression of *FOS*, a transcription factor relevant to many physiological and pathological processes, but also significant overexpression of keratin (*KRT*) genes and bone morphogenetic protein 2 gene (*BMP2*) in DSLD [3]. The RNA-seq data can be found at https://www.ncbi.nlm.nih.gov/Traces/study/?acc=PRJNA544650. The *BMP2* finding correlated well with detection of BMP2 in whorl-like cellular formation of active DSLD lesions or foci as reported by us [5].

We hypothesized that the examination of tissues other than tendon is more feasible as DSLD is a systemic disorder not confined to tendons/ligaments. To test our hypothesis we selected a series of targets for diagnosis of DSLD that were found among overexpressed genes in skin transcriptomes in our previous work [3] and utilized immunohistochemistry (IHC) and a sophisticated version of in situ hybridization (RNAscope assay) to see that these targets can be utilized as DSLD biomarkers in the skin.

## Methods and materials

### Experimental subjects

Two groups of horses were used. Each group consisted of control and DSLD-affected horses with a slight overlap of subjects between the two groups. Group 1 was used in RT-qPCR assay and group 2 was utilized for IHC and RNAscope assays (Tables 1 and 2). Skin biopsies from the neck area under the mane were collected from all horses in both groups. Diagnosis of DSLD was confirmed through post-mortem examination of tendons and/or ligaments at necropsy or through clinical examination. Collection of skin biopsies from living horses and tissues at necropsies was approved by The University of Georgia IACUC (AUP # A2022 06-037-Y1-A0).

Skin biopsies obtained from the first group of horses were immersed in RNALater solution and used for RNA extraction. Several samples of RNAs or rather cDNAs were saved from samples used for RNA-seq as reported by us previously [3]. All these samples were used for quantitative real-time PCR (RT-qPCR) (Table 1).

Another set of skin biopsies was obtained from 24 DSLD-diagnosed horses and 14 control horses. The horses varied in age, breed, and gender (Table 2). Several skin samples were collected from horses donated to College of Veterinary Medicine at The University of Georgia or from horses coming to necropsy at the Department of Pathology. Many biopsies and tissues collected post mortem were submitted by veterinarians from the US and overseas. Several skin samples were collected from horses that also provided skin biopsies for RNA extraction in a previous study (Table 1) [3]. All skin samples were obtained from the posterior neck area under the mane, immersed in formalin and/or in RNALater (Qiagen, Germantown, MD) and embedded in paraffin. The histological sections from these blocks were used for immunohistochemistry and RNAscope assays to evaluate the presence and level expression of several potential markers.

### Real time quantitative PCR (RT-qPCR)

RT-qPCR was used to validate the overexpression of several genes in each horse [3]. We used differences in the transcriptomes between the controls and DSLD horses to indicate which overexpressed genes would be most suitable to test as biomarkers to diagnose DSLD. RNAs for several potential biomarkers were run through RT-qPCR assays and evaluated based on average Cq scores. Gene expression level (fold difference) was calculated with $2^{-\Delta\Delta Ct}$ method using glyceraldehyde-3-phosphate dehydrogenase (GAPDH) and actin beta (ACTB) as reference genes [6].

Total RNA was extracted using RNAeasy Mini Kit following manufacturer's instructions. (QIAGEN) and quantified using a Nanodrop 2000 spectrophotometer (ThermoFisher Scientific, Waltham, MA, USA). RNA quality was confirmed by checking that A260/A280 ratios were 2.0±0.1 for all samples tested, then the total RNA concentration was further determined by Qubit 3.0 Fluorometer (ThermoFisher Scientific, Suwanee, GA, USA).

RT-qPCR was carried out on a CFX96 Real -time system (Bio-Rad Laboratories, Inc., Hercules, CA) in two steps. First, synthesis of cDNA using iScript Advanced cDNA synthesis Kit

**Table 1. Experimental subjects providing RNA samples for RT-qPCR.**

| Diagnosis | Sex | Age (years) | Breed | Diagnostics |
|---|---|---|---|---|
| DSLD I | F | 30 | QH | Necropsy |
| DSLD II | F | 13 | Santa Cruz | Necropsy |
| DSLD III | M | 23 | WB | Clinical |
| DSLD IV | M | Late 20's | WB | Clinical |
| DSLD V | M | 30 | TB | Clinical |
| DSLD VI | F | 22 | WB | Clinical |
| DSLD VII | F | 27 | TB | Necropsy |
| DSLD VIII | F | 21 | WB | Clinical |
| DSLD IX | M | 5 | Hanoverian | Necropsy |
| DSLD X | F | 30 | Arabian | Necropsy |
| DSLD XI | M | 15 | WB | Necropsy |
| DSLD XII | F | 11 | Paint WB cross | Necropsy |
| DSLD XIII | F | 12 | Hanoverian | Necropsy |
| DSLD+ | M | 20 | PP | Necropsy |
| DSLD+ | F | 5 | PP | Necropsy |
| DSLD+ | F | mid 20's | PP | Necropsy |
| DSLD+ | M | 18 | WB | Clinical |
| DSLD+ | M | 15 | PP | Clinical |
| DSLD+ | M | mid 20's | WB TB | Necropsy |
| DSLD* | M | 16 | PP | Clinical |
| DSLD* | M | 8 | PP | Clinical |
| Control I | M | N/A | WB | Clinical |
| Control II | M | 2 | Holsteiner | Clinical |
| Control III | | 18 | Danish WB | Clinical |
| Control IV | M | 25 | Dutch WB | Clinical |
| Control V | | 23 | Hanoverian | Clinical |
| Control VI | M | 24 | WB | Necropsy |
| Control VII | | 22 | Oldenburg | Clinical |
| Control VIII | | 27 | Hanoverian | Clinical |
| Control+ | F | 22 | PP | Necropsy |
| Control+ | F | 23 | PP | Necropsy |
| Control+ | F | 22 | PP | Necropsy |
| Control+ | F | 29 | PP | Clinical |
| Control+ | M | 13 | PP | Clinical |
| Control+ | F | 3 | PP | Clinical |
| Control*9 | F | 26 | PP | Clinical |
| Control*10 | F | 8 | PP | Clinical |
| Control*11 | F | 5 | PP | Clinical |
| Control*12 | F | 3 | PP | Clinical |

+ original RNA samples used in RNA-seq study [3].

*additional skin biopsies used for IHC and RNAscope assays.

Necropsy: this includes cases where full necropsy was done or only a few pieces of a tendon/ligament were available.

PP: Peruvian Paso breed.

WB: Warmblood.

TB: Thoroughbred.

**Table 2. Experimental and control subjects used in IHC and RNAscope.**

| Description | Sex | Age | Breed | Diagnostic method |
|---|---|---|---|---|
| Control 1 | M | 24 | Tenn Walking | Necropsy |
| Control 2 | M | 13 | Appaloosa | Necropsy |
| Control 3 | M | 9 | Oldenberg | Necropsy |
| Control 4 | M | 27 | Thoroughbred | Necropsy |
| Control 5 | F | 5 | Quarter | Necropsy |
| Control 6 | M | n/a | Quarter | Necropsy |
| Control 7 | M | n/a | Quarter | Necropsy |
| Control 8 | M | n/a | Quarter | Necropsy |
| Control 9* | F | 26 | Peruvian Paso | Clinical |
| Control 10* | F | 8 | Peruvian Paso | Clinical |
| Control 11* | F | 8 | Peruvian Paso | Clinical |
| Control 12* | F | 6 | Peruvian Paso | Clinical |
| Control 13 | M | 10 | Arabian | Clinical |
| Control 14 | M | 24 | Arabian | Clinical |
| DSLD 1 | F | 7 | Peruvian Paso | Necropsy |
| DSLD 2 | M | 13 | Gray Tenn Walk | Clinical |
| DSLD 3* | M | 13 | Peruvian Paso | Clinical |
| DSLD 4* | M | 8 | Peruvian Paso | Clinical |
| DSLD 5 | M | 11 | WB | Necropsy |
| DSLD 6 | M | 12 | Andalusian | Clinical |
| DSLD 7 | F | 14 | German Riding Pony | Necropsy |
| DSLD 8 | F | 16 | Arabian | Necropsy |
| DSLD 9 | F | 10 | Arabian | Clinical |
| DSLD 10 | F | 11 | Arabian | Clinical |
| DSLD 11 | M | 6 | Peruvian Paso | Clinical |
| DSLD 12 | M | 20 | Quarter | Necropsy |
| DSLD 15 | M | 27 | Paint TB | Necropsy |
| DSLD 16 | M | 10 | Spanish Breed | Clinical |
| DSLD 17 | M | 14 | Zweibrucker | Necropsy |
| DSLD 18 | F | 17 | Arabian | Clinical |
| DSLD 19 | F | 17 | Arabian | Clinical |
| DSLD 20 | F | 14 | Arabian | Clinical |
| DSLD 21 | F | 8 | Arabian | Clinical |
| DSLD 22 | M | 16 | German WB | Clinical |
| DSLD 23 | M | 11 | Warmblood | Clinical |
| DSLD 24 | M | 10 | Peruvian Paso | Clinical |
| DSLD 25 | F | 26 | Thoroughbred | Necropsy |
| DSLD 26+ | M | 6 | Holsteiner | Necropsy |

*Additional skin biopsies from these horses were used for RNA extraction for RT-qPCR (Table 1).

Necropsy: this includes cases where full necropsy was done or only a few pieces of a tendon/ligament were available

+Used for RNAscope assay only

(Bio-Rad), according to the manufacturer's instructions was performed. Briefly, a 20μl reaction containing 2μg of total RNA was used to synthesize cDNA as follows: 46°C for 20 min and 95°C for 1min. The RT-qPCR was carried out in a 20μl reaction containing 2μl of 20-fold diluted cDNA using SsoAdvanced Universal SYBR Green Supermix (Bio-Rad) with the

**Table 3. Primers, their sequences and target.**

| Primer name | Sequence (5'-3') | Target | Gene ID |
|---|---|---|---|
| H-KRT83-F | TGTTGGATCTGTGAATGTCTGTGT | Equus caballus keratin 83 (KRT83) | 100146665 |
| H-KRT83-R | CAGGATCCGGCAGTGGTGTT | | |
| H-KRT81-F | CCAAGAGCCAGAACTCCAAG | Equus caballus keratin81(KRT81) | 100062930 |
| H-KRT81-R | GTAGGTGGCGATCTCGATGT | | |
| H-KRT39-F | GACCCGCCACAGTCTAACAT | Equus caballus keratin39(KRT39) | 100067401 |
| H-KRT39-R | CATAGCGGGTCTCCGTCTCC | | |
| BMP2-F | GCGGTCTCCTAAAGGTCCTC | Equus caballus bone morphogenetic protein 2 (BMP2) | 100051701 |
| BMP2-R | CAACTCGAACTCGCTCAGGA | | |
| FOS-F | GGTCAATGCGCAGGACTACT | Equus caballus (FOS) | 001491972 |
| FOS-R | TACCGAGGAGACCAGGGTG | | |
| qH-GAPDH-F | GTTTGTGATGGGCGTGAACC | Equus caballus glyceraldehyde-3-phosphate dehydrogenase (GAPDH) | 001163856.1 |
| qH-GAPDH-R | TGCACTGTGGTCATGAGTCC | | |
| qH-ACT-F | ATGATGATATCGCCGCGCTC | Equus caballus actin beta (ACTB), mRNA | 100033478 |
| qH-ACT-R | CCACCATCACGCCCTGG | | |

conditions as follows: 95°C for 30sec, followed by 40 cycles of 2-step PCR (94°C denaturing for 15 sec, 60°C annealing for 30 sec), then 95°C 10 sec and Melt curve 60°C to 95°C with increment of 0.5°C. The selection of specific targets for RT-qPCR primers was based on target expression in transcriptomes in DSLD horses as reported in our RNA-seq study [3]. These primers were constructed using NCBI ExPrimer program to design primers from exon—exon junctions (Table 3).

## Immunohistochemistry

Five-micron thick histological sections of formalin-fixed paraffin-embedded tissues (FFPET) were cut onto positively charged slides (Probe On Plus, Fisher Scientific, Springfield, NJ) and deparaffinized. The slides were blocked by Peroxidazed 1 (cat # PX968, Biocare Medical, Pacheco, CA, USA) for 5 min. Antigen retrieval was accomplished by microwaving in citrate buffer solution (Antigen Unmasking Solution, Vector Laboratories, Burlingame, CA). Background protein was blocked by Background Punisher (cat # BP974, Biocare) for 10min. This was followed by application of one of the primary antibodies and incubated overnight at 4°C. The following primary rabbit polyclonal antibodies were used:

- anti-keratin 39 (cat # AP32987SU, OriGene Technologies, Inc., Austin, TX, USA),

- anti-keratin 81 (cat # AA 142–257, https://www.antibodies-online.com/productsheets/),

- anti-keratin 83 (cat # orb184603, Biorbyt, Cambridge, UK),

- anti-BMP2 (cat # PA1-31215, ThermoFisher Scientific),

- anti-FOS (cat # ab209794, Abcam, Boston, MA, USA) antibodies.

The concentrations for each antibody were: 1:300 for anti-KRT39, 1:300 for anti- KRT81, 1:500 for anti-KRT83 (1:500), and 1:300 for both anti- BMP2 and anti -FOS antibodies.

Biotin-free technology was used to detect rabbit polyclonal primary antibodies, followed by an alkaline phosphatase (AP) polymer that binds to the probe (MACH 3 Rabbit AP-Polymer Detection System (cat # M3R533L, Biocare Medical). The slides were first incubated for 10 minutes at room temperature with MACH 3 Rabbit Probe then for another 10 minutes at RT with MACH 3 Rabbit AP-Polymer. Chromogen was applied for 5–7 min at room temperature

using Biocare Warp Red. (cat # WR806S, Biocare Medical). To avoid interference in staining between heavy presence of melanin in horse skins and the more commonly used DAB, red stain protocol was used for both IHC and RNAscope ISH (see below). Slides were counterstained with 50% hematoxylin and coverslipped.

We evaluated the intensity of immunostaining of several histological features of the skin, including hair follicle, epidermis, and/or sebaceous glands on a scale of 0–4, with a level of 0 indicating no expression and a level 4 indicating strong overexpression.

## RNAscope in-situ hybridization

Next, we performed RNAscope *in-situ* hybridization on 5μm thick histological sections from the same blocks used from the IHC study to evaluate the gene expression of each potential biomarker in the skin. All samples were processed using the RNAscope 2.5 HD Detection Kit (red) as recommended by the manufacturer (Advanced Cell Diagnostics [7], Inc., Newark, CA) with six hybridization amplifications. Our horse probes for each gene were custom designed by ACD with each probe containing 20 double-Z probe pairs (Each target probe contained an 18- to 25-base region complementary to the target RNA, a spacer sequence, and a 14-base tail sequence (conceptualized as Z). A pair of target probes (double Z), each possessing a different type of tail sequence, hybridized contiguously to a target region) covering about 1000bp length. Each probe was designed according to the corresponding gene sequence: RNAscope Probe®Ec-*KRT39*-C1 [*Equus caballus* keratin 39 (*KRT39*), mRNA NM_001346158.1); RNAscope Probe®Ec-*KRT81*-C1 [*Equus caballus* keratin, type II cuticular Hb1 (*KRT81*), mRNA NM_001346216.1], RNAscope Probe®Ec-*FOS* [*Equus caballus* Fos proto-oncogene, AP-1 transcription factor subunit (*FOS*), mRNA XM_001491972.4], RNAscope Probe®Ec-*BMP2*-C2 [*Equus caballus BMP2* transcript variant X1 mRNA XM_001493895.5], RNAscope Probe ®Ec-*KRT83*-C1, [*Equus caballus* keratin 83 (*KRT83*) mRNA NM 001346211)].

We tested the accuracy of the technique through the use of positive and negative controls (provided by ACD) which were effective in ensuring maximum quality and accuracy of our results.

Five micron thick tissue sections were deparaffinized in xylene, followed by dehydration in an ethanol series. Tissue sections were pretreated with hydrogen peroxide for 10min at room temperature, rinsed with deionized water, then maintained in retreatment solution at a boiling temperature (100–103˚C) using a hot plate for 15 minutes, rinsed in deionized water, and immediately treated with protease at 40˚C for 30 minutes in a HybEZ hybridization oven (ACD, Hayward, CA). After pretreatment, tissue sections were incubated with desired probe for 2 hours at 40˚C.

Amplification and detection steps were performed with the RNAscope 2.0 HD Detection Kit reagents (cat # ACD 322360) for single-plex probes. The sections were incubated with Amp1 for 30 min at 40˚C, with Amp2 for 15 min at 40˚C, with Amp3 for 30 min at 40˚C and with Amp4 for 15 min at 40˚C. Then the slides were incubated with Amp5 for 30min at RT, with Amp6 for 15 min at RT, between each incubation steps the slides were washed with washing buffer twice. Signal was detected by fast Red (1:60 diluted with Red A buffer) for 10min at RT, the slides were counterstained with 50% hematoxylin for 2min at RT. The dried slides were permanently cover-slipped.

The number of positive follicles per 25 follicles present was used to determine positivity vs negativity for the respective biomarker. From this data, the percent positive value was calculated using the formula (# affected/total follicle count of 100). In addition, the intensity of *FOS* expression in the epidermis was determined to be absent, mildly moderately or strongly expressed. Absent meant that no *FOS* expression was detected, moderately expressed meant

that there were small punctate and/or intermittent patterns of expression throughout the epidermis, particularly along the basal side, and significantly expressed meant that the majority of the epidermis was expressing *FOS* (greater than 80% of the entire epidermis)

## Statistics

A two-sided T-test as well as the Wilcoxon rank sum test were used to compare the results of immunohistochemistry and RNAscope assays for each biomarker along with the mean, standard deviation, and percent difference between control and DSLD affected horses (JMP Pro 2022 Statistical Program). We used a *p* value of less than 0.05 to determine statistical significance, and biomarkers determined significant through these tests were established as the best representatives for a panel that may be used to diagnose DSLD. For non-statistically significant results (P>0.05), a power analysis was conducted ($\alpha$ = 0.05, two-sided T-test) to determine the risk of committing Type II errors design.

Using JMP Pro 16, receiver operating characteristic (ROC) curves graphically displaying the trade-off between sensitivity and specificity were generated for different cut-off points. For the diagnostic assay to adequately distinguish between normal/controls (X axis) and diseased/ DSLD horses /Y axis), only ROC curves with an area under the curve (AUC) of 0.7 or higher were considered as considered. We selected our cut off value using Youden's Index. We also considered one value above and one value below Youden's Index to include varying levels of sensitivity and specificity.

## Results

### Experimental subjects and skin sample collection

As noted in Methods, skin samples were collected from two groups: one group of 18 control and 21 DSLD horses was used in RT-qPCR assay (Table 1). A second group of 14 control horses for IHC and RNAscope assays and 23 DSLD-affected horses was utilized. Twenty four DSLD-affected horses and 14 control horses were included in RNAscope assays (Table 2). DSLD presence or absence was established in some cases by clinical examination of limbs and ultrasound, and by necropsy in others (Tables 1 and 2). To determine which location for skin biopsy would be most appropriate and would give most consistent results in IHC and RNAscope assays six different body sites were used for skin biopsy in a preliminary experiment to confirm the overexpression of the genes was uniform regardless of the biopsy site. Since DSLD is a systemic disorder, we expected that there would be little to no difference in expression of the selected genes in any of the sampling skin sites. All selected biomarkers were detectable in all six sites and the expression of each studied gene and presence of corresponding antigens was comparable in each site. The posterior neck skin under the mane was chosen for the standard biopsy site so this would result in less aesthetic damage as scar tissue and healing of the skin would be minimally noticeable and could easily be covered by the mane. These considerations are particularly important for show horses or athletes.

### RT- qPCR and biomarker selection

Previous data from our RNA-seq study showed differential expression in many genes in skin, including *BMP2*, *FOS*, *KRT39*, *KRT81*, and *KRT83* [3]. RT-qPCR was used to validate the overexpression of *FOS*, *BMP2*, some of keratin genes and other genes in each horse [3]. We used differences in the transcriptomes between the controls and DSLD horses to indicate which overexpressed genes would be most suitable to test as biomarkers to diagnose DSLD. RNAs for several potential biomarkers were run through RT-qPCR assays and evaluated based on

**Table 4. RT-qPCR results.** RT-qPCR average Cq for 20 DSLD horse skin samples and 18 control horse skin samples. The Cq values were average of total 20 DSLD horse skin samples and 18 normal horse skin samples. Each sample was run twice in triplicates. Results showed there was a larger fold difference between DSLD and control for *KRT83* and *KRT81* (with Cq value 24–27 and 26–27, respectively).

| Target | DSLD Average Cq | Normal Average Cq | Fold difference (DSLD value to Normal value) $2^{-\Delta\Delta Cq}$ * | |
|---|---|---|---|---|
| | | | GAPDH as reference gene | B-Actin as reference gene |
| *KRT81* | 26.5 | 27.31 | 2.14 | 1.98 |
| *KRT83* | 30.34 | 35.28 | 10.41 | 9.98 |
| *KRT31* | 24.84 | 27.7 | 3.25 | 3.10 |
| *KRT36* | 32.25 | 35.53 | 4.55 | 4.08 |
| *KRT39* | 33.09 | 33.31 | 1.33 | 1.03 |
| *KRT40* | 30.68 | 33.65 | 2.80 | 2.71 |
| hyaluronan synthase 3 (*HAS3*) | 36.66 | 38.7 | 1.9 | 1.76 |
| cell migration inducing hyaluronan binding protein (*CEMIP4*) | 30.38 | 30.55 | 0.8 | 0.8 |
| *BMP2* | 34.84 | 37.61 | 4.81 | 4.68 |
| *FOS* | 27.46 | 29.41 | 1.064 | 1.01 |

*$2^{-\Delta\Delta Cq}$ means: *KRT81* $2^{-\Delta\Delta Cq}$ equal to 2.14 means DSLD horses are expressing *KRT81* at a 2.14 higher level than normal horses

$2^{-\Delta\Delta Cq}$ equal to 1 means there is no difference at gene expressing level

average Cq scores (Table 4). *KRT81* $2^{-\Delta\Delta Cq}$ was equal to 2.14. This indicated that DSLD horses were expressing *KRT81* at a 2.14 higher level than the normal horses. $2^{-\Delta\Delta Cq}$ equal to 1 means there was no difference at gene expression level. *KRT83* was also overexpressed in DSLD samples. *KRT39* and *KRT81* genes were significantly overexpressed in DSLD horses. In addition, the RNA expression level of *BMP2* was tested as well. Though the Cq difference between DSLD (Cq 34.84) and control (Cq 37.61) *BMP2* was not significant, we nevertheless included *BMP2* among potential biomarkers, because of its presence in active tendon lesions in DSLD [5] (Fig 1). Likewise the Cq difference in *FOS* expression between DSLD and control samples was not significant, we nevertheless looked for *FOS* expression using IHC and RNAscope, because RNA-seq indicated that *FOS* the most overexpressed gene in DSLD [3]. We performed RT-qPCR for several more *KRT*s (*KRT31*, *36* and *40*), and for *CEMIP4* and *HAS3* genes encoding for cell migration inducing hyaluronan binding protein and hyaluronan 3, respectively. However, as the Cq differences were not significant these genes were not included for further consideration as biomarkers (Table 4).

## Immunohistochemistry

We evaluated the intensity of immunostaining of several histological features of the skin, including hair follicle, epidermis, and/or sebaceous glands for the presence of each biomarkers —keratins 39, 81 and 83, BMP2, and FOS (Table 5, Fig 2).

Keratin 81 presence was higher in controls than in DSLD horses. The intensity of staining was 2.2 in control hair follicles and 2.4 in control sebaceous glands, but only 1.4 in DSLD hair follicles and 1.6 in DSLD sebaceous glands (P<0.05 for both structures, respectively, Fig 2, Table 5).

Keratin 39 presence was markedly higher in controls than in DSLD with a mean staining intensity of 2.4 in the follicles and 2.3 in the sebaceous glands, while in DSLD cases the mean staining intensity in the follicles was 1.8 and in the sebaceous glands it was 1.9. We determined that Keratin 39 overall was only marginally significant (P = 0.06; 0.8 power n = 15; 0.9 power n = 20) (Fig 2).

Keratin 83, unlike KRT39 and KRT81, had higher presence in DSLD horses and lower presence in controls for all horses in which KRT83 was detected. Keratin 83 mean staining

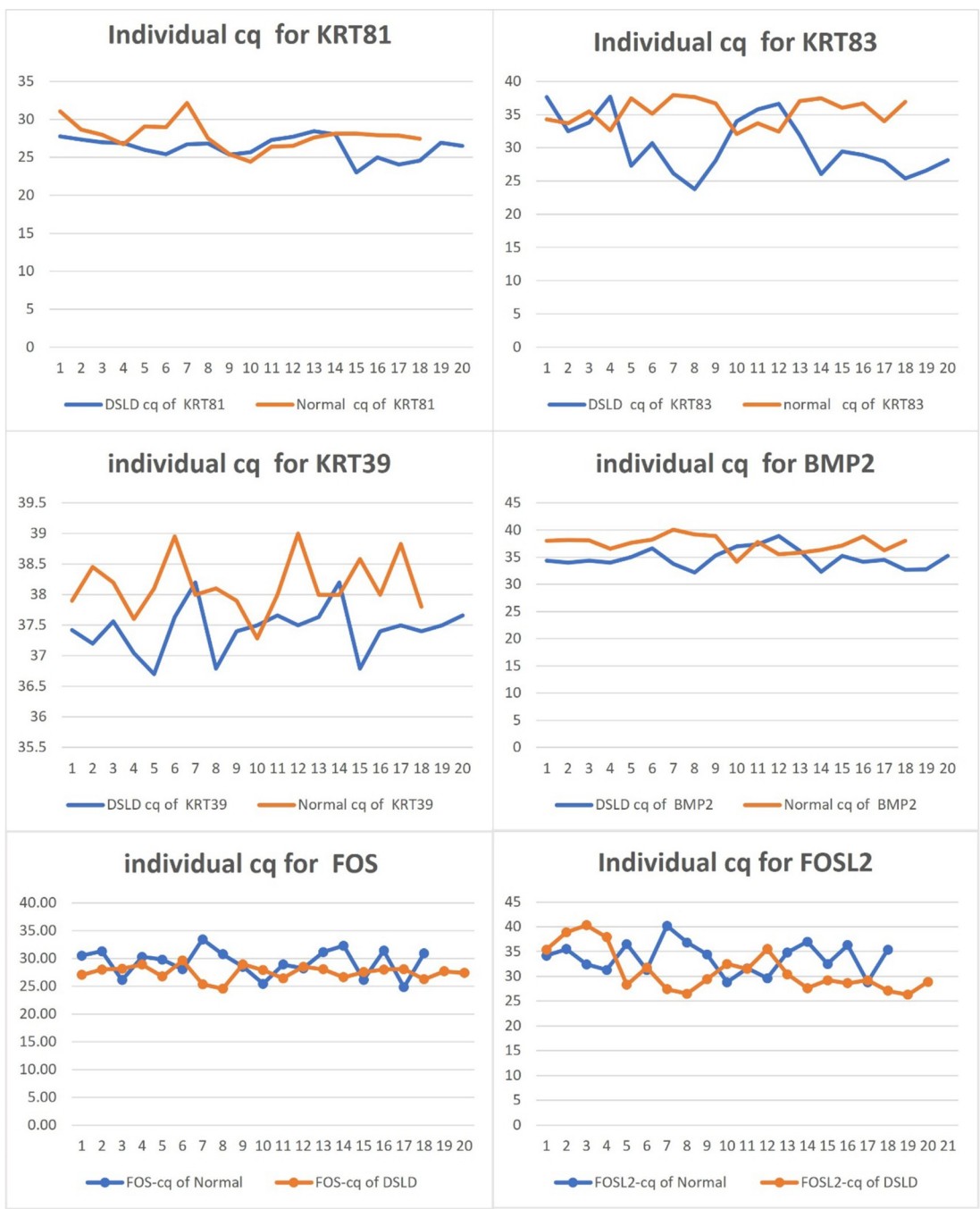

**Fig 1. RT-qPCR results for selected individual biomarkers.** Blue—DSLD Cq for each biomarker. Orange—control Cq for each biomarker.

intensity was 2.6 in follicles and 2.3 in DSLD horses, but only 1.5 and 1.6 in follicles and sebaceous glands of controls, respectively (P<0.05 for both regions) (Fig 2).

BMP2 was also present at higher level in horses with DSLD when compared with controls. The mean staining intensity for follicles and sebaceous glands in DSLD horses was 1.75 for follicles and 2.0 for sebaceous glands, while in control horses it was 0.6 for follicles and 0.6 for sebaceous glands (P<0.05 for both regions combined) (Fig 2).

**Table 5. Immunohistochemistry data.**

| Description | Sex | Age | Breed | K39 IHC (Fol) | K39 IHC (Seb) | K81 IHC (Fol) | K81 IHC (Seb) | K83 IHC (Fol) | K83 IHC (Seb) | BMP2 IHC (Fol) | BMP2 IHC (Seb) | FOS IHC |
|---|---|---|---|---|---|---|---|---|---|---|---|---|
| Control 1 | M | 24 | Tenn Walking | 3 | 3 | 3 | 3 | 3 | 3 | 0.5 | 2 | 0 |
| Control 2 | M | 13 | Appaloosa | 3 | 3 | 2.5 | 3 | 1 | 1 | 1 | 1 | 3 |
| Control 3 | M | 9 | Oldenberg | 3 | 1 | 2.5 | 3 | 1 | 1 | 1.5 | 2 | 1 |
| Control 4 | M | 27 | Thoroughbred | 2 | 2 | 2.5 | 3 | 3 | 4 | 1.5 | 2 | 0 |
| Control 5 | F | 5 | Quarter | 3 | 3 | 3 | 2.5 | 0 | 1 | 1 | 1 | 2 |
| Control 6 | M | n/a | Quarter | 2 | 3 | 2 | 2 | 2 | 2 | 0 | 0 | 2 |
| Control 7 | M | n/a | Quarter | 2 | 1 | 2.5 | 3 | 3 | 4 | 0 | 0 | 0 |
| Control 8 | M | n/a | Quarter | 2 | 2 | 2 | 2 | 0 | 0 | 2 | 1 | 2 |
| Control 9 | F | 26 | Peruvian Paso | 3 | 3 | 0 | 0 | 4 | 0 | 0 | 0 | 2.5 |
| Control 10 | F | 8 | Peruvian Paso | 2 | 3 | 2 | 2 | 0 | 1 | 0 | 0 | 2.5 |
| Control 11 | F | 8 | Peruvian Paso | 2 | 2 | 1 | 0 | 1 | 1 | 0 | 0 | 1 |
| Control 12 | F | 6 | Peruvian Paso | 2 | 2 | 0 | 1 | 1 | 2 | 0 | 0 | 3 |
| Control 13 | n/a | n/a | Arabian | 3 | 3 | 1 | 2 | 1.5 | 1.5 | 1 | 1 | 2 |
| Control 14 | n/a | n/a | Arabian | 2 | 1 | 0 | 0 | 1 | 2 | 0 | 0 | 0 |
| DSLD 1 | F | 7 | Peruvian Paso | 2 | 1 | 0 | 1 | 3 | 3 | 2 | 1 | 2.5 |
| DSLD 2 | M | 13 | Gray Tenn Walk | 2 | 2 | 2.5 | 3 | 0 | 1 | 3 | 3 | 1 |
| DSLD 3 | M | 13 | Peruvian Paso | 1.5 | 3 | 0 | 0 | 0 | 4 | 0 | 2 | 4 |
| DSLD 4 | M | 8 | Peruvian Paso | 3 | 3 | 0 | 0 | 0 | 1 | 0 | 0 | 3 |
| DSLD 5 | N/A | N/A | Peruvian Paso | 2 | 2 | 0 | 0 | 1 | 3 | 0.5 | 0 | 2.5 |
| DSLD 6 | M | 12 | Andalusian | 3 | 2 | 3 | 3 | 0 | 1 | 1 | 2 | 2 |
| DSLD 7 | F | 14 | German Riding Pony | 2 | 2 | 3 | 3 | 3 | 3 | 0 | 0 | NSA |
| DSLD 8 | F | 16 | Arabian | 1 | 1 | 1 | 3 | 2 | 1 | 2 | 2 | NSA |
| DSLD 9 | F | 10 | Arabian | 3 | 3 | 3 | 3 | 1 | 1 | 1 | 2 | 3 |
| DSLD 10 | F | 11 | Arabian | 2 | 2 | 3 | 3 | 2 | 0 | 0 | 2 | 3 |
| DSLD 11 | M | 6 | Peruvian Paso | 2 | 1 | 2.5 | 3 | 0 | 0 | 3 | 3 | 3 |
| DSLD 12 | M | 20 | Quarter | 3 | 3 | 3 | 3 | 2 | 3 | 3 | 3 | 3 |
| DSLD 15 | M | 27 | Painted TB | 2 | 2 | 2 | 3 | 3 | 3 | 0 | 0 | 1.5 |
| DSLD 16 | M | 10 | Spanish Breed | 1 | 2 | 0 | 0 | 3 | 3 | 0 | 0 | 1.5 |
| DSLD 17 | M | 14 | Zweibrucker | 3 | 2 | 3 | 3 | 3 | 3 | 2 | 2 | 2 |
| DSLD 18 | F | 17 | Arabian | 2 | 1 | 2 | 1 | 1 | 1 | 1.5 | 1.5 | 3 |
| DSLD 19 | F | n/a | Arabian | 1.5 | 2 | 1 | 1 | 3 | 3 | 1 | 1 | 2 |
| DSLD 20 | F | n/a | Arabian | 2 | 3 | 1 | 1 | 3 | 2 | 0 | 0 | 0 |
| DSLD 21 | F | n/a | Arabian | 2 | 2 | 2 | 2 | 3 | 3 | 1 | 1 | 2 |
| DSLD 22 | M | 15 | German WB | 0.5 | 0 | 1 | 1 | 3 | 3 | 0 | 0 | 0 |
| DSLD 23 | M | 11 | Thoroughbred | 1 | 1.5 | 0 | 0 | 3 | 1 | 0 | 0 | 3 |
| DSLD 24 | M | 10 | Peruvian Paso | 0.5 | 0 | 0 | 0 | 4 | 4 | 0 | 0 | 0 |
| DSLD 25 | F | 26 | Thoroughbred | 2 | 2 | 0 | 0 | 3 | 3 | 0 | 0 | 3 |

NSA = no sample available.

Last, FOS was present to a higher degree in horses with DSLD but less in the controls. Since FOS was not present in the sebaceous glands, the samples were analyzed for overall FOS presentation and not generalized to a specific histological structure region of skin, including epidermis and hair follicles. The mean staining intensity of FOS for DSLD horses was 2.5 and the

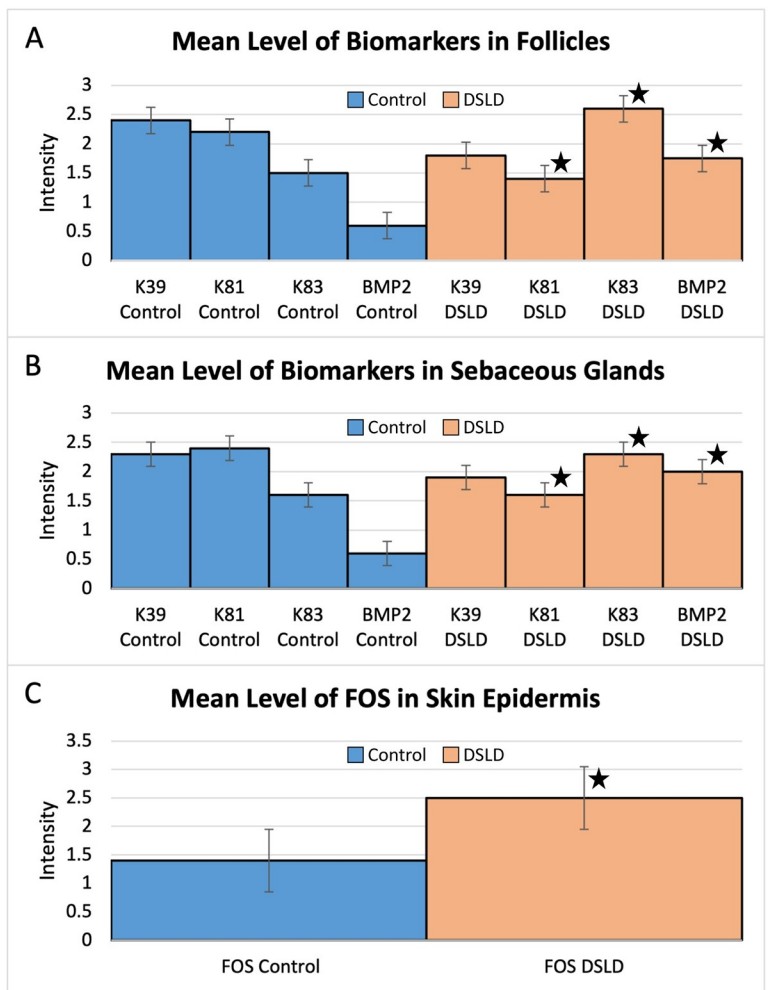

| D Biomarker | Mean Control | Mean DSLD | Percent Difference | P Value |
|---|---|---|---|---|
| K39 Follicle | 2.4 | 1.8 | -33 | >0.06 |
| K39 Sebaceous | 2.3 | 1.9 | -21 | >0.05 |
| K81 Follicle | 2.2 | 1.4 | -57 | >0.05 |
| K81 Sebaceous | 2.4 | 1.6 | -50 | >0.05 |
| K83 Follicle | 1.5 | 2.6 | 73 | >0.05 |
| K83 Sebaceous | 1.6 | 2.3 | 44 | >0.05 |
| BMP2 Follicle | 0.6 | 1.75 | 66 | >0.05 |
| BMP2 Sebaceous | 0.6 | 2 | 70 | >0.05 |
| FOS Skin | 1.4 | 2.5 | 79 | >0.05 |

Minus percent difference indicates that DSLD samples contained less biomarker than control.

**Fig 2. Summary of immunohistochemical staining for all markers.** A. Mean staining of each biomarker in follicles. (K39 control = mean 2.4, p = 0.06, 0.8 power n = 15, 0.9 power n = 20, p<0.05; K81 control = mean 2.2, p<0.05; K83 control = mean 2.6, p<0.05; BMP2 control = mean 0.6, p<0.05. K39 DSLD = 1.8, p = 0.06, 0.8 power n = 15, 0.9 power n = 20; K81 DSLD = mean 1.4, p<0.05; K83 DSLD = mean 2.6 p<0.05; BMP2 DSLD = mean 1.75, p<0.05). Star = highly significant difference between DSLD and corresponding control value. B. Mean staining of each biomarker in sebaceous glands. (K39 control = mean 2.3, p = 0.06, 0.8 power n = 15, 0.9 power n = 20, p<0.05; K81 control = mean 2.4, p<0.05; K83 control = mean 1.6, p<0.05; BMP2 control = mean 0.6, p<0.05. K39 DSLD = 1.9, p = 0.06, 0.8 power n = 15, 0.9 power n = 20; K81 DSLD = mean 1.6, p<0.05; K83 DSLD = mean 2.3 p<0.05; BMP2 DSLD = mean 2.0, p<0.05). C. Mean of FOS in horses with DSLD compared with controls. Control = 1.4 and DSLD = 2.5 (P<0.01). D. Immunohistochemistry staining parameters such as mean intensity, P-values, and percent differences between controls and DSLD cases or each biomarker. P values calculated using Wilcoxon rank sum test through JMP program.

mean staining intensity for control horses was 1.4 for samples in which FOS was detected, and was found to be highly significant (P<0.01). (Fig 2).

Table 5 shows detailed results for each biomarker and each experimental subject. In summary, we found no significant data to demonstrate that there were any significant differences in the presence of these markers in breed or gender that may have affected the results.

## RNAscope

**Age dependent presence and overexpression of biomarkers.** Our RNAscope assay readily detected the five biomarkers in 14 control and 24 DSLD horses (Table 6). The results revealed a significant increase in several of our selected biomarkers in horses with DSLD, and even more in older horses with DSLD. The average lifespan of a horse is 25 to 30 years, depending on the breed, and our study included horses as young as <1–5 years and as old as 25 years. Older horses with DSLD (aged 20+ years) had noticeably higher expression of several of the biomarkers when compared with control horses in the same age group (Fig 3C and 3D). The expression of *FOS* was highest in DSLD horses aged 16–20 years. The expression of keratin 39 increased in older horses as detected by RNAscope, but not with IHC in both control and DSLD groups (Fig 3). RNAscope ISH showed almost no difference in biomarker detection in *KRT83* and *BMP2* in the controls, and in some cases, *FOS* was not detected at all in older controls (Fig 3C). Because the number of horses in each age group was limited the statistical evaluation was not done for results presented in Fig 3.

**Table 6. RNAscope data.** Scoring for each biomarker (except for *FOS* expression in epidermis) was done by percentage. *FOS* epidermis score was based on intensity 0 = none present and +++ = high intensity. N/A = not available.

| RNAscope Control | | | | | | | | RNAscope DSLD | | | | | | |
|---|---|---|---|---|---|---|---|---|---|---|---|---|---|---|
| Case | K39 | K81 | K83 | BMP2 | FOS | FOS Epidermis | | Case | K39 | K81 | K83 | BMP2 | FOS | FOS Epidermis |
| Control 1 | 76 | 0 | 56 | 0 | 0 | 0 | | DSLD 1 | 30 | 10 | 0 | 50 | 100 | +++ |
| Control 2 | 28 | 100 | 8 | 0 | 0 | 0 | | DSLD 2 | 0 | 20 | 0 | 20 | 100 | 0 |
| Control 3 | 100 | 10 | 0 | 0 | 0 | 0 | | DSLD 3 | 25 | 0 | 14 | 12 | 0 | 0 |
| Control 4 | 0 | 0 | 36 | 29 | 0 | 0 | | DSLD 4 | N/A | N/A | 100 | 80 | 80 | +++ |
| Control 5 | 100 | 10 | 58 | 0 | 100 | 0 | | DSLD 5 | 100 | 100 | 80 | 73 | 0 | +++ |
| Control 6 | 71 | 30 | 36 | 80 | 30 | +++ | | DSLD 6 | 67 | 100 | 20 | 0 | 0 | ++ |
| Control 7 | 60 | 30 | 10 | 34 | 100 | +++ | | DSLD 7 | 20 | 0 | 75 | 0 | 0 | + |
| Control 8 | 100 | 0 | 32 | 60 | 50 | 0 | | DSLD 8 | 0 | 20 | 0 | 50 | 0 | 0 |
| Control 9 | 100 | 0 | 0 | 0 | 0 | 0 | | DSLD 9 | 0 | 100 | 59 | 0 | 0 | ++ |
| Control 10 | 0 | 20 | 25 | 0 | 50 | ++ | | DSLD 10 | 60 | 0 | 10 | 80 | 100 | +++ |
| Control 11 | 0 | 67 | 10 | 0 | 0 | 0 | | DSLD 11 | 100 | 0 | 0 | 0 | 100 | + |
| Control 12 | 100 | 30 | 50 | 10 | 0 | 0 | | DSLD 12 | 0 | 50 | 77 | 100 | 95 | ++ |
| Control 13 | 10 | 80 | 30 | 20 | 0 | 0 | | DSLD 15 | 0 | 100 | 0 | 100 | 100 | +++ |
| Control 14 | 90 | 50 | 20 | 50 | 0 | 0 | | DSLD 16 | 50 | 80 | 100 | 100 | N/A | +++ |
| | | | | | | | | DSLD 17 | 100 | 100 | 76 | 75 | 100 | +++ |
| | | | | | | | | DSLD 18 | 50 | N/A | 10 | 100 | 100 | +++ |
| | | | | | | | | DSLD 19 | 100 | 100 | 100 | 50 | 0 | ++ |
| | | | | | | | | DSLD 20 | 76 | 100 | 95 | 86 | 100 | +++ |
| | | | | | | | | DSLD 21 | 0 | 100 | 0 | 100 | 100 | 0 |
| | | | | | | | | DSLD 22 | 10 | 10 | 0 | 0 | 100 | 0 |
| | | | | | | | | DSLD 23 | 10 | 100 | 70 | 0 | 100 | ++ |
| | | | | | | | | DSLD 24 | 80 | 100 | 30 | 10 | 0 | ++ |
| | | | | | | | | DSLD 25 | 75 | 80 | 50 | 0 | 0 | +++ |
| | | | | | | | | DSLD 26 | 100 | 0 | 100 | 100 | 100 | +++ |

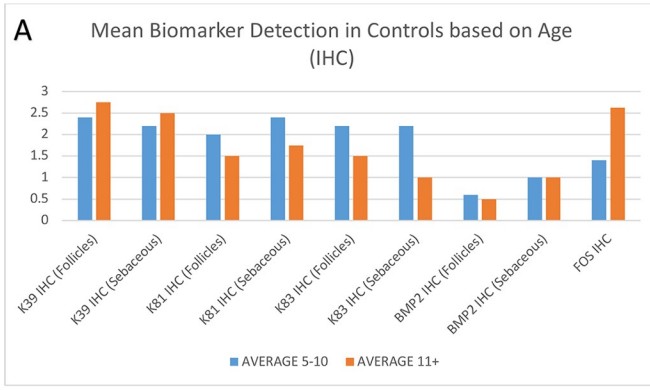

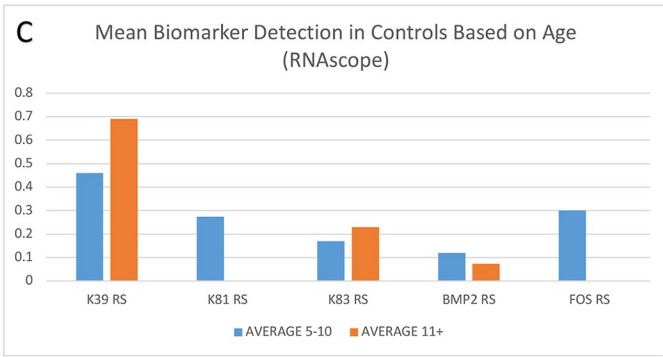

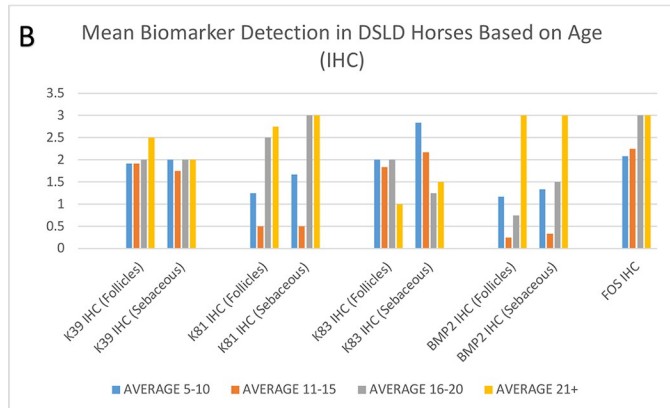

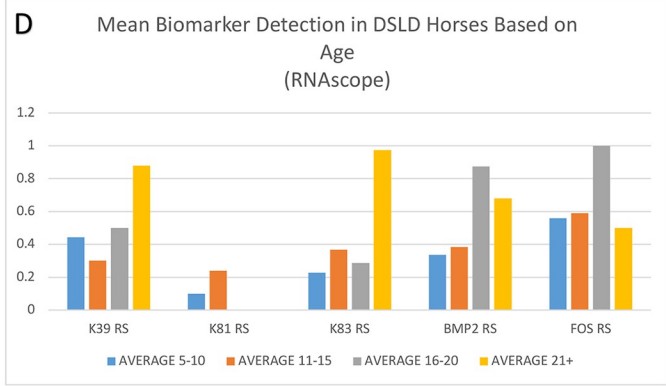

**Fig 3. Mean detection of each biomarker in horses based on age.** A. Mean biomarker detection through IHC based on horse age (controls only). Blue bar = ages 5–10 years. Orange bar = any horse over age 11. B. Mean biomarker detection through IHC based on age (DSLD only) in 5 year intervals. C. Mean biomarker age through RNAscope based on horse age (controls only). Blue bar = ages 5–10 years. Orange bar = any horse over age 11. D. Mean biomarker detection through RNAscope based on age (DSLD only) in 5 year intervals.

RNAscope assay confirmed the overexpression data obtained by RT-qPCR for *KRT81*, *KRT83*, *BMP2*, and *FOS* in DSLD skin (Table 4). The expression of all biomarkers was increased in cases of DSLD with the exception of *KRT39* that was actually lower in horses with DSLD (Table 6, Figs 4 and 5). The intensity of *FOS* expression in the DSLD epidermis is visualized in Fig 5.

Each biomarker was assessed using the Wilcoxon rank sum test, and was marked significant for P<0.05 (Table 7). *FOS* showed the most impressive difference in expression. *FOS* was overexpressed in DSLD, and the number of follicles expressing *FOS* was far less in controls (P<0.001; Sd 0.069). *BMP2* was expressed at a higher percentage in DSLD cases when compared with controls (P = 0.033, Sd = 0.15). *KRT83* also was expressed at a significantly higher percentage in DSLD cases (P = 0.037; Sd = 0.11). *KRT39* was the only biomarker that was found in a higher percentage of follicles in controls when compared to DSLD and that was considered significant (P = 0.045; Sd = 0.15). *KRT81* did not show a statistically significant difference in expression between controls and DSLD cases.

Power analysis was conducted to determine if the number of samples collected would yield statistical significance of 80% power. Sensitivity and specificity was calculated using JMP 16 to generate an ROC curve in which AUC equal to or greater than 0.70 was considered significant (Table 6). Our results show that *FOS* and *BMP2* RNAscope values were significant (AUC = 70). The cut off value for *FOS* expression was 0.80 with a 75% chance of the horse having DSLD. Sensitivity of the *FOS* test was 62% and specificity of the *FOS* test was 86%. For

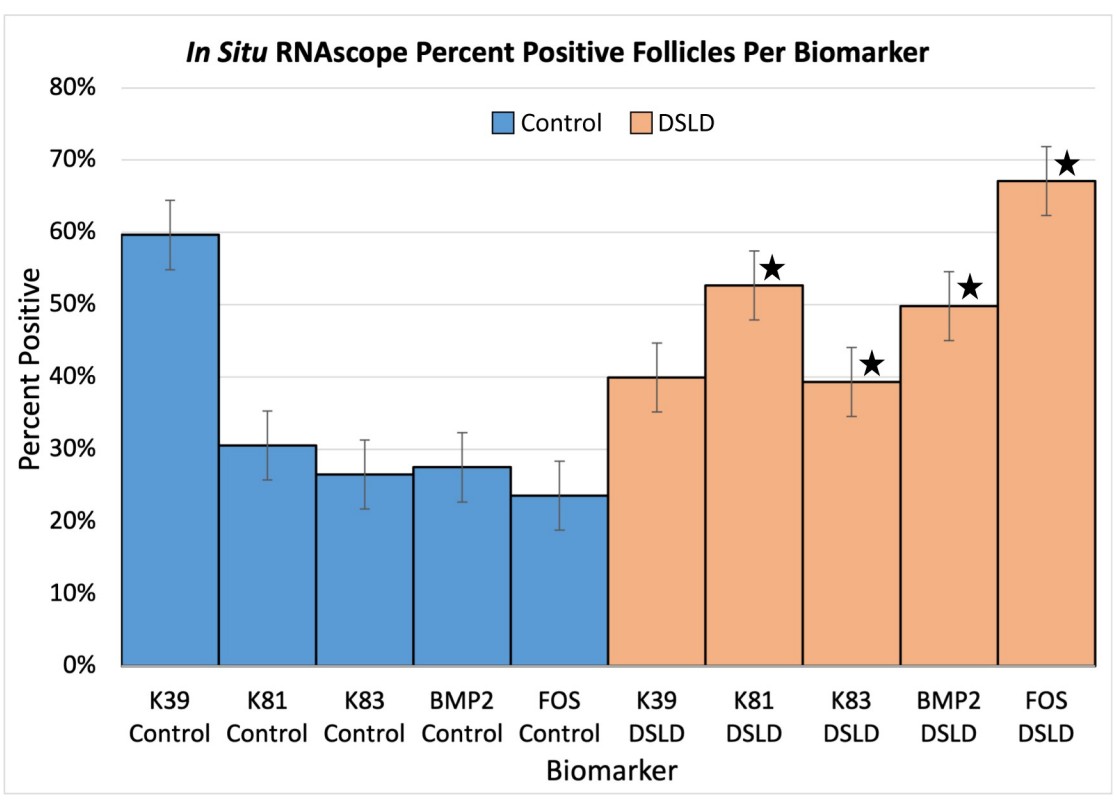

**Fig 4. Percent positive follicles by biomarker for control and DSLD cases.**

*BMP2* expression the cut off value was 0.73 with a 71% chance of the horse having DSLD. The sensitivity of the *BMP2* test was 43%, and the specificity was 88%. However, each of the three keratins, including *KRT83* had an AUC under 0.70 and so the calculated sensitivity for these indices was too low and specificity was too high (Table 8).

## Criteria for selection of potential biomarkers and diagnostic panel establishment

Though the immunohistochemical staining was different between control and DSLD skin biopsies, it was not statistically significant for any of the tested biomarkers. In addition, there was significant variation among individuals with DSLD. However, RNAscope assays gave direction which indices have the potential to serve as biomarkers for DSLD. Quantification was based on percentage of stained hair follicles for keratin genes and *BMP2*, or in the case of *FOS* on percentage of stained epidermis.

Our data show that *FOS*, *BMP2* and *KRT83* were all readily detectable by RNAscope in skin of horses with DSLD at a significantly higher level than that of controls, regardless of breed or gender. In addition, the expression of *BMP2* and *FOS* were detected at higher levels in horses between the ages of 16–20 years, specifically significantly higher in DSLD horses than in controls, leading us to include both of them in our panel. Though *KRT83* was present in significantly more follicles in DSLD horses than in controls as well, its AUC level under 0.70 with resulting low sensitivity and high specificity indicates that *KRT83* is not suitable for use as a biomarker unless future studies involving more horses show otherwise. At the present time, *FOS* and *BMP2* provide the best option to serve as biomarkers of DSLD with optimal specificity and sensitivity based on Youden's Index.

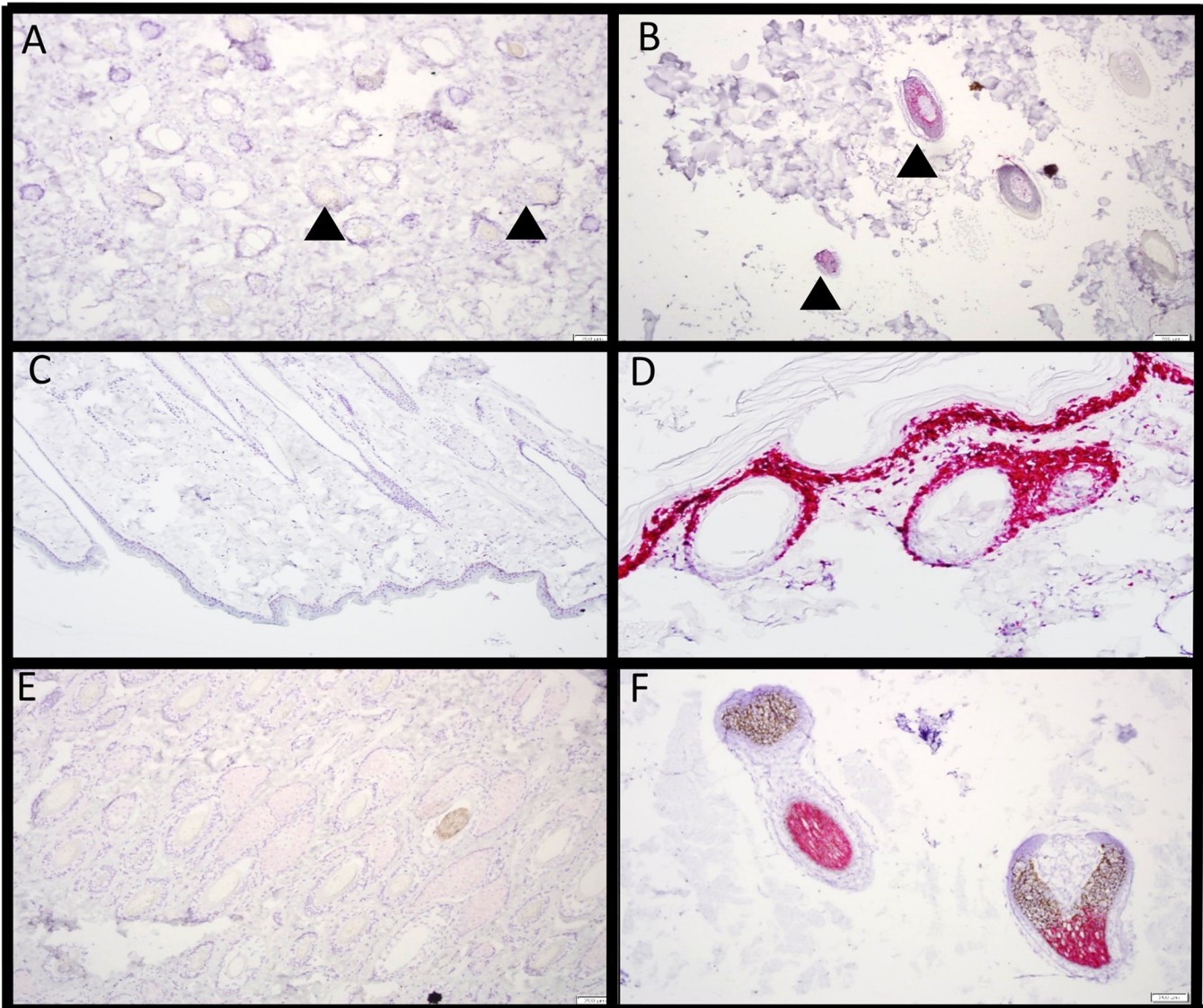

**Fig 5. *In situ* hybridization with RNAscope.** A. *BMP2* RNAscope in control skin, arrowheads point to negative hair follicles. B. *BMP2* RNAscope in DSLD skin, arrowheads point to mild to moderately positive hair follicles. C. *FOS* RNAscope in control skin. D. *FOS* RNAscope in DSLD skin. E. *KRT81* RNAscope in control skin. F. *KRT81* RNAscope in DSLD skin. All, 200 x magnification.

**Table 7. Quantification of RNAscope *in-situ* hybridization values.** Mean, P-values, and percent difference between controls and DSLD cases were determined for each biomarker. P values were calculated using Wilcoxon rank sum test through JMP program.

| Biomarker | Avg Control | Avg DSLD | Percent Difference | P Value | Standard Dev. |
|---|---|---|---|---|---|
| *KRT39* | 0.596 | 0.399 | -40% | P = 0.046 | 0.16 |
| *KRT81* | 0.305 | 0.526 | 53% | P>0.05 | 0.15 |
| *KRT83* | 0.265 | 0.393 | 39% | P = 0.039 | 0.12 |
| *BMP2* | 0.275 | 0.498 | 58% | P = 0.035 | 0.17 |
| *FOS* | 0.236 | 0.671 | 96% | P<0.001 | 0.071 |

**Table 8. Sensitivity and specificity of biomarkers for DSLD.**

| Biomarker | X (cut off value) | Probability of having DSLD | Sensitivity | Specificity | True Pos | True Neg | False Pos | False Neg | AUC |
|---|---|---|---|---|---|---|---|---|---|
| FOS | 0.95 | 0.80 | 0.57 | 0.86 | 12 | 12 | 2 | 9 | 0.70 |
| FOS | 0.80* | 0.75 | 0.62 | 0.86 | 13 | 12 | 2 | 8 | |
| FOS | 0.50 | 0.63 | 0.62 | 0.71 | 13 | 10 | 4 | 8 | |
| BMP2 | 0.75 | 0.71 | 0.39 | 0.875 | 9 | 14 | 2 | 14 | 0.70 |
| BMP2 | 0.73* | 0.71 | 0.43 | 0.875 | 10 | 14 | 2 | 13 | |
| BMP2 | 0.65 | 0.68 | 0.43 | 0.81 | 10 | 13 | 3 | 13 | |
| KRT81 | 1.0 | 0.79 | 0.41 | 0.93 | 9 | 13 | 1 | 13 | 0.63 |
| KRT81 | 0.8* | 0.73 | 0.5 | 0.86 | 11 | 12 | 2 | 11 | |
| KRT81 | 0.67 | 0.69 | 0.5 | 0.79 | 11 | 11 | 3 | 11 | |
| KRT83 | 0.70 | 0.74 | 0.40 | 1 | 9 | 14 | 0 | 14 | 0.57 |
| KRT83 | 0.59* | 0.71 | 0.43 | 1 | 10 | 14 | 0 | 13 | |
| KRT83 | 0.58 | 0.70 | 0.43 | 0.93 | 10 | 13 | 1 | 13 | |
| KRT39 | 0.71 | 0.58 | 0.65 | 0.5 | 15 | 7 | 7 | 8 | 0.60 |
| KRT39 | 0.67* | 0.59 | 0.65 | 0.57 | 15 | 8 | 6 | 8 | |
| KRT39 | 0.60 | 0.60 | 0.61 | 0.57 | 14 | 8 | 6 | 8 | |

*Maximized Youden's Index.

## Discussion

This is the first study that has attempted to use overexpression of targets in skin as biomarkers for a systemic disorder severely affecting tendon and ligaments in horses. Horses with DSLD were shown to have changes in expression of several markers, with statistically significant overexpression of BMP2 and FOS in skin in various structures within the skin and hair follicles. It is unclear if the increased expression is a result of a mutation of any of these genes, or a consequence of a mutation in another gene yet to be identified, or a result of another genetic alteration, such as single nucleotide polymorphism [8]. Type 2 keratin genes are closely placed in clusters on one or two chromosomes in several species [9], while BMP2, FOS, and other keratins are localized on completely different chromosomes. Further studies are required to analyze the genome of the DSLD horse to determine if it is indeed a heritable genetic disorder that can be detected through innovative standardized laboratory tests.

Though skin pathology is not the major sign of DSLD, skin changes have been noticed by owners of DSLD horses [3]. This rather less documented or even an obscure finding has been an impetus to utilize skin biopsies for DSLD diagnosis. Using RNAscope ISH we found gene overexpression of KRT81, KRT83, BMP2 and FOS in skin of horses with DSLD, and used this set to develop a diagnostic panel for DSLD. Other studies have come out in the past decade describing the use of biomarkers found in skin to diagnose major diseases. In fact, two very recent studies on amyotrophic lateral sclerosis (ALS) demonstrated that TDP-43 was a significant biomarker for the disease, despite the fact that ALS does not result in significant or observable skin conditions [10, 11]. New studies have been investigating whether skin disorders common in Parkinson patients could be used as an aid to diagnose Parkinson disease where diagnosis is based on clinical findings as there are no known biomarkers for Parkinson [12, 13]. A 2020 study by Showalter et al. assembled a panel of biomarkers found in skin that were representative of systemic sclerosis and could potentially be used to stage disease progression [14]. Kidney disorders have also taken a leap into the possibility of diagnosis through biomarkers present in skin [15]. Similarly, we propose a novel diagnostic panel that identifies elevated levels of BMP2 and FOS as biomarkers of DSLD.

Interestingly, this study revealed also an increase in expression of *KRT81*, *KRT83*, *BMP2* and *FOS* genes with aging in control and DSLD horses, though this increase was much more significant in the DSLD group. This novel discovery not only supports the notion that DSLD is a chronic systemic disorder that gets worse over time, but may also lead to a more accurate diagnostic plan that involves follow up of affected or suspected horses for the overexpression of these biomarkers over time to establish disease severity and progression. We can only speculate whether similar, but much less substantial changes observed in the control group indicate relationship between "degenerative changes" of DSLD and aging. This novel finding could mean that a reliable diagnostic assay for DSLD would need to account for age differences based on onset of symptoms and age of the horse, perhaps through a grading system that measures overexpression at key milestones of the horse's life (such as achievement of breeding age), or on yearly intervals.

We have no explanation why RNA-seq, but not RNAscope (as both assays were performed on skin biopsies from control and DSLD horses) identified overexpression of *KRT39* in skin from DSLD horses. The lower presence of KRT39 in DSLD by IHC can be understood as consequence of decreased or impaired translation of *KRT39* mRNA.

Through our sensitivity and specificity analysis, we determined that *in-situ* hybridization through RNAscope yielded the most accurate and significant data. RNAscope technique has become a reliable method to evaluate gene expression in clinical diagnostic tests [16]. In our study we found that a cut off value of 0.80 was most effective for *FOS*, while 0.73 was an effective cut off for *BMP2*. We determined that the normal range of the biomarkers should be less than 50% for *FOS*, and less than 65% in *BMP2*, as these values yielded too many false negatives (8 for *FOS* and 13 for *BMP2*, n = 22). However, though overexpression of *KRT83* appeared to be statistically significant, its AUC was the lowest from our tested markers (0.57) and it did not perform well with regard to specificity and sensitivity (Table 8).

Though the overexpression of *BMP2* in RNA-seq was only marginally significant, the inclusion of *BMP2* in this study was warranted because of previous documentation of increased presence of BMP2 in active foci of fibroblasts/tenoblasts in DSLD tendons and ligaments in our lab [5]. We consider these foci to be early lesions of DSLD that lead to inappropriate accumulation of certain connective tissue components, such as proteoglycans over time, resulting in chronic symptoms that worsen over the affected horse's lifetime. This correlates well with the data achieved through our biomarkers, which also accumulate far more dramatically with age in affected horses than in controls. Our work demonstrated that DSLD is a systemic disorder resulting in accumulation of proteoglycans [1, 4] but it also results in the accumulation of other structural proteins such as keratin, and other types of proteins such as BMP-2 and FOS [3, 5]. It is also apparent that as the disease progresses and as the horse ages that these biomarkers do accumulate, possibly in connection with worsening of symptoms and disorder progression over time.

While immunohistochemistry was helpful in identifying presence of proteins encoded by these three genes, we found that *in situ* hybridization was more accurate and consistent, and easier to read. In particular, RNAscope technology is advantageous as it is much easier to identify expression. There is less background from staining and processing that could lead to a false positive staining which was problematic with IHC. It also revealed punctate points of expression which were easier to distinguish in and around individual cells in the tissue when compared to immunohistochemistry.

Our results clearly show a significant change of expression in these genes that were identified as overexpressed in our RNA-seq study. We recommend further study of each of these biomarkers with a larger sample pool to confirm that a panel of the two biomarkers (*FOS*, *BMP2*) can be suggested to veterinarians as a useful tool in diagnosing DSLD in mature horses

suspected of having DSLD. It might be used for follow up of young horses with family history of DSLD before they achieve breeding age. On a yearly basis. Information and understanding of DSLD is still at its infancy, and it is very possible that there are many horses who are carriers for DSLD or too young to present with clear DSLD symptoms to be categorized by veterinarians as carrying a pathognomonic abnormal DSLD conformation and being lame. They may present with marginal conformational modifications or with reduced performance at the start, not clear enough for veterinarians to suspect an emerging DSLD. Whether this panel can be used as a prospective tool to identify DSLD in such horses as they age remains to be seen. We also recommend to continue RNA scope testing for *KRT83* though *KRT83* had an AUC under 0.70 with resulting sensitivity too low and specificity too high, the p value of 0.039 in RNA scope assays is significant enough to warrant to test more horses.

One drawback of our study is that the diagnosis of DSLD was based on clinical examination and ultrasound in some subjects. It is difficult to obtain enough necropsy specimens and we really appreciate the owners and veterinarians who contributed samples to our study. We anticipate that this paper will encourage more owners and veterinarians to send us skin biopsies so we can expand our work and confirm the diagnostic value of our panel.

## Acknowledgments

We would like to thank the following horse owners and veterinarians for contributing and/or obtaining horse tissue samples to this study: Marianne Bowman, Mary Barbara Alexander, Drs. Lauren Schnabel, Valerie Moorman, Ellen Korsgaard, Michael Zager, Moira Nusbaum, Cooper Williams, Darla K. Moser, Brittany Williamson, and Braidee Foote. Dr. Sheila Keys provided invaluable guidance with statistical evaluation.

## Author Contributions

**Conceptualization:** Jaroslava Halper.

**Data curation:** Jennifer Hope Roberts, Jian Zhang, Amy McLean, Karen Blumenshine.

**Formal analysis:** Jennifer Hope Roberts, Florent David, Jaroslava Halper.

**Funding acquisition:** Jennifer Hope Roberts, Karen Blumenshine, Jaroslava Halper.

**Investigation:** Jennifer Hope Roberts, Jian Zhang, Amy McLean, Eva Müller-Alander.

**Methodology:** Jennifer Hope Roberts, Jian Zhang, Florent David, Amy McLean, Karen Blumenshine, Jaroslava Halper.

**Project administration:** Jaroslava Halper.

**Resources:** Amy McLean, Karen Blumenshine, Eva Müller-Alander.

**Software:** Jennifer Hope Roberts.

**Supervision:** Jaroslava Halper.

**Validation:** Jennifer Hope Roberts, Jaroslava Halper.

**Visualization:** Jennifer Hope Roberts.

**Writing – original draft:** Jennifer Hope Roberts.

**Writing – review & editing:** Jennifer Hope Roberts, Jian Zhang, Karen Blumenshine, Jaroslava Halper.

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
