## [Decision Letter · Decision Letter 0]

2 May 2023

PONE-D-23-01721Expression of genes with biomarker potential identified in skin from DSLD-affected horses increases with agePLOS ONE

Dear Dr. Halper,

Thank you for submitting your manuscript to PLOS ONE. After careful consideration, we feel that it has merit but does not fully meet PLOS ONE’s publication criteria as it currently stands. Therefore, we invite you to submit a revised version of the manuscript that addresses the points raised during the review process.

We look forward to receiving your revised manuscript.

Kind regards,

Sivaraj Mohana Sundaram

Academic Editor

PLOS ONE

3. We noted in your submission details that a portion of your manuscript may have been presented or published elsewhere. [Data identifying several overexpressed genes was used for development. The data was taken from our previous publication: Haythorn, A., et al., Differential gene expression in skin RNA of horses affected with degenerative suspensory ligament desmitis. Journal of Orthopaedic Surgery and Research, 2020. 15(1): p. 460.] Please clarify whether this publication was peer-reviewed and formally published. If this work was previously peer-reviewed and published, in the cover letter please provide the reason that this work does not constitute dual publication and should be included in the current manuscript.

Additional Editor Comments:

The presentation of the results needs to be improved. Please include all statistical significance test methods, dF, and exact p-values in the results section.

Bar graphs require individual data points, standard deviations, and graph color codes that should be defined within each figure.

The quality of scale bars on immunostaining pictures should be improved with a readable text size.

The findings and outlook could be summarized in an abstract figure.   

Reviewers' comments:

Reviewer's Responses to Questions

**Comments to the Author**

1. Is the manuscript technically sound, and do the data support the conclusions?

Reviewer #1: Yes

2. Has the statistical analysis been performed appropriately and rigorously? 

Reviewer #1: Yes

3. Have the authors made all data underlying the findings in their manuscript fully available?

Reviewer #1: Yes

4. Is the manuscript presented in an intelligible fashion and written in standard English?

Reviewer #1: Yes

5. Review Comments to the Author

Reviewer #1: Abstract: well written, no comments.

Introduction: This work is based on the results from the authors previous work. They explain their previous work but I feel, they need to mention their horse breeds used in that study in introduction. Is there any other NGS study performed? Other than this group? This was not mentioned. Also their previous study was a transcriptomics study. NGS is a borad term to use.

Material and methods: In line 142, RT-PCR, please add reference for 2-ΔΔCt method. In line 160, Table 3, is GAPDH is only reference gene? This work depends highly on RT-PCR, so adding one more reference gene or area will be better.

Results: They are bit too elaboarted but understandable. Figures are without standard deviation/ standrad error bars. They need to be inserted.

Discussion: In line 351, Was NGS study performed on RNA isolated from skin tissues of affected DSLD horses?

Finally, I like the way authors recognises their drawbacks.

6. PLOS authors have the option to publish the peer review history of their article (what does this mean?). If published, this will include your full peer review and any attached files.

Reviewer #1: No

---

## [Author Response · Author response to Decision Letter 0]

30 May 2023

Concerning certain Journal requirements, we did provide information regarding animal treatment (in the Supporting Information section). The study was approved by institutional IACUC (line 108 in Material and Methods) Details about submission of supporting data from a previously done RNA-seq (Haythorn et al., 2020) can be found in an NIH repository at https://www.ncbi.nlm.nih.gov/Traces/study/?acc=PRJNA544650. Yes, the paper by Haythorn et al was published, see ref. number 3 in the manuscript. It is not a dual publication to this manuscript under review as it reports RNA -seq and identification of differentially expressed genes in RNA-seq. There was no attempt to establish any diagnostic criteria or assays using those data in the paper by Haythorn et al. The URL for the NCBI entry is now included in Data Availability section and in the manuscript itself (line 91).

The phrase “data not shown” was omitted. The reference list is complete and up to date. 

Response to Editor comments: Statistical test methods and exact p-values are now included in the Result section and/or Figure legends where appropriate. Statistical evaluation was not done for results in Figure 3 because as we noted in the text the number of horses in each age group was limited (lines 365 and 366). We do not have scale bars on Fig. 5 (immunohistochemistry), but it is stated in the Figure legend that all photos are taken at 200x magnification. We did put together a Graphic abstract.

Response to Reviewer’s comments: 

No. 3: https://www.ncbi.nlm.nih.gov/Traces/study/?acc=PRJNA544650 is included in the body of the paper in the Introduction (line 91), and in Data Availability at the end of the manuscript. : Statistical test methods and exact p-values are included in the Result section and/or Figure legends. 

No. 5: NGS was changed to RNA-seq throughout the manuscript. Equine geneticists are more interested in SNP analysis where breeds do play a role. Because we do not know whether DSLD is actually a single entity, and because Peruvian Paso horses suffer from higher prevalence of DSLD we cannot exclude the possibility that breed plays a role. The breed importance is comparable to the role of ethnicity and race background in human disease. 

Introduction: We did add that in addition to Peruvian Pasos we were able to obtain skin biopsies from several other breeds, e,g, Arabians, Warmbloods and Hanoverians (lines 70 and 71). As far as we know no other RNA-seq study has been done so far (line 87).

Material and methods: A reference for 2-∆∆Ct method using glyceraldehyde-3-phosphate dehydrogenase (GAPDH) and actin beta (ACTB) was added (ref 6) as was one more reference gene (actin beta): line 146, Tables 3 and 4).

Results: standard deviations and statistical differences were marked in appropriate figures.

Discussion: yes, RNA-seq was performed on RNA isolated from skin samples from control and DSLD horses (lines 472 and 473), and also see in Introduction, lines 87 and 88.

---

## [Decision Letter · Decision Letter 1]

13 Jun 2023

Expression of genes with biomarker potential identified in skin from DSLD-affected horses increases with age

PONE-D-23-01721R1

Dear Dr. Halper,

We’re pleased to inform you that your manuscript has been judged scientifically suitable for publication and will be formally accepted for publication once it meets all outstanding technical requirements.

Kind regards,

Sivaraj Mohana Sundaram

Academic Editor

PLOS ONE

Additional Editor Comments (optional):

Reviewers' comments:

Reviewer's Responses to Questions

**Comments to the Author**

1. If the authors have adequately addressed your comments raised in a previous round of review and you feel that this manuscript is now acceptable for publication, you may indicate that here to bypass the “Comments to the Author” section, enter your conflict of interest statement in the “Confidential to Editor” section, and submit your "Accept" recommendation.

Reviewer #1: All comments have been addressed

2. Is the manuscript technically sound, and do the data support the conclusions?

Reviewer #1: Yes

3. Has the statistical analysis been performed appropriately and rigorously? 

Reviewer #1: Yes

4. Have the authors made all data underlying the findings in their manuscript fully available?

Reviewer #1: Yes

5. Is the manuscript presented in an intelligible fashion and written in standard English?

Reviewer #1: Yes

6. Review Comments to the Author

Reviewer #1: I have no comments this time. Authors answered comments and I accept the manuscript. Authors worked well on previous comments.

7. PLOS authors have the option to publish the peer review history of their article (what does this mean?). If published, this will include your full peer review and any attached files.

Reviewer #1: No

---

## [Editor Report · Acceptance letter]

5 Jul 2023

PONE-D-23-01721R1 

*Expression of genes with biomarker potential identified in skin from DSLD-affected horses increases with age*

Dear Dr. Halper:

I'm pleased to inform you that your manuscript has been deemed suitable for publication in PLOS ONE. Congratulations! Your manuscript is now with our production department. 

Kind regards, 

on behalf of

Dr. Sivaraj Mohana Sundaram 

Academic Editor

PLOS ONE